# Higher pNRF2, SOCS3, IRF3, and RIG1 Tissue Protein Expression in NASH Patients versus NAFL Patients: pNRF2 Expression Is Concomitantly Associated with Elevated Fasting Glucose Levels

**DOI:** 10.3390/jpm13071152

**Published:** 2023-07-18

**Authors:** Suzan Schwertheim, Malek Alhardan, Paul P. Manka, Jan-Peter Sowa, Ali Canbay, Hartmut H.-J. Schmidt, Hideo A. Baba, Julia Kälsch

**Affiliations:** 1Department of Gastroenterology, Hepatology and Transplant Medicine, University Hospital of Essen, University of Duisburg-Essen, 45147 Essen, Germany; suzan.schwertheim@gmail.com (S.S.); malek.alhardan@uk-essen.de (M.A.); hartmut.schmidt@uk-essen.de (H.H.-J.S.); 2Institute of Pathology, University Hospital of Essen, University of Duisburg-Essen, 45147 Essen, Germany; hideo.baba@uk-essen.de; 3Department of Medicine, Ruhr University Bochum, University Hospital Knappschaftskrankenhaus Bochum, 44892 Bochum, Germany; paul.manka@rub.de (P.P.M.); jan.sowa@rub.de (J.-P.S.); ali.canbay@rub.de (A.C.)

**Keywords:** NASH, pNRF2, SOCS3, immunohistochemistry, liver disease

## Abstract

Non-alcoholic fatty liver disease (NAFLD) embraces simple steatosis in non-alcoholic fatty liver (NAFL) to advanced non-alcoholic steatohepatitis (NASH) associated with inflammation, fibrosis, and cirrhosis. NAFLD patients often have metabolic syndrome and high risks of cardiovascular and liver-related mortality. Our aim was to clarify which proteins play a role in the progression of NAFL to NASH. The study investigates paraffin-embedded samples of 22 NAFL and 33 NASH patients. To detect potential candidates, samples were analyzed by immunohistochemistry for the proteins involved in innate immune regulation, autophagy, apoptosis, and antioxidant defense: IRF3, RIG-1, SOCS3, pSTAT3, STX17, SGLT2, Ki67, M30, Caspase 3, and pNRF2. The expression of pNRF2 immunopositive nuclei and SOCS3 cytoplasmic staining were higher in NASH than in NAFL (*p* = 0.001); pNRF2 was associated with elevated fasting glucose levels. SOCS3 immunopositivity correlated positively with RIG1 (r = 0.765; *p* = 0.001). Further, in NASH bile ducts showed stronger IRF3 immunostaining than in NAFL (*p* = 0.002); immunopositive RIG1 tissue was higher in NASH than in NAFL (*p* = 0.01). Our results indicate that pNRF2, SOCS3, IRF3, and RIG1 are involved in hepatic lipid metabolism. We suggest that they may be suitable for further studies to assess their potential as therapeutics.

## 1. Introduction

NAFLD is a worldwide increasing problem, and patients often have metabolic syndrome and high risks of cardiovascular and liver-related mortality. Recently, it was documented that constitutive active innate immune signaling can lead to excessive inflammatory cytokine release and consequently can promote the development of hepatic steatosis and fibrosis [1]. Nevertheless, the factors leading to the progression of NAFL to NASH are still unknown. Much of the liver research relates to outcomes in mouse models, but data on protein expression levels specifically in NAFLD patients on this topic are rare. Therefore, we asked if proteins of innate immunity are deregulated in our NAFLD group and if there is an association with the clinical data of our patients. Among others, pattern recognition receptors (PRRs) participate in the regulation of lipid metabolism. Extracellular pathogens or endogenous injury signals are initially detected by PRRs on cell membranes or endosomal membranes [1]. Retinoic acid-inducible gene 1 (RIG1/DDX58) belongs to a family of cytosolic pattern recognition receptors (PRR) and triggers an innate immune response. Briefly, the pathogenic association molecule pattern (PAMP) is recognized by RIG1 and activates mitochondrial antiviral signaling protein (MAVS)-dependent signals which lead to activation of IRF3 and NF-κB and subsequent production of type I/II IFN and inflammatory cytokines [1]. Among others, the proinflammatory cytokine IL-6 is activated by RIG1 and IL-6 activates the JAK-STAT signaling pathway. Based on our results, we assume signaling cascades activated by PRRs, FFA accumulation, and ROS formation (for more details see Figure 1). The signal transducer and activator of transcription 3 (STAT3) belong to the STAT family of cytoplasmic transcription factors [2]. STAT3 is to a large extent known as an oncogenic factor in various human cancers [2]. The suppressor of cytokine signaling 3 (SOCS3) proteins is also known as STAT-induced STAT inhibitor (SSI3) [3]. It has been documented that in obesity SOCS3 is upregulated in concert with increases in inflammation in the hypothalamus, adipose tissue, and liver [4]. The transcription factor nuclear factor-erythroid 2 related factor 2 (NRF2) is also known to participate in hepatic fatty acid metabolism [5] and to regulate the innate immune response [6]. NASH has been shown to elicit lipid peroxidation, accumulation of reactive oxygen species (ROS), and proinflammatory cytokines in the liver, which leads to liver injury and inflammation [7]. Exposure to oxidative stress and inflammatory conditions activates NRF2 inducing the expression of cytoprotective genes [8]. Furthermore, the involvement of autophagy in hepatocyte lipid metabolism has recently been demonstrated [9,10]. Impairment of autophagic flux is closely associated with NAFLD [9,11,12]. Therefore, we analyzed our patient group on the implication of autophagy by using Syntaxin (STX17), a SNARE protein, formerly successfully used in liver tissue for autophagy detection [13]. STX17 translocates to autophagosomes and mediates the fusion of autophagosomes with lysosomes which enables the degradation of autophagosome contents [14]. Briefly, we selected these proteins for our current studies based on previous study interests and staining availability. Further, a major risk factor for non-alcoholic steatohepatitis (NASH) is insulin resistance with elevated blood glucose [15]. Sodium–glucose cotransporter 2 (SGLT2) is the major cotransporter known to participate in glucose reabsorption in the kidney. In a mouse model with diabetes, NASH/cirrhosis/HCC SGLT2 expression was detected in liver tumors [16]. Recently, the SGLT2 inhibitor NGI001 inhibited diet-induced metabolic dysfunction and non-alcoholic fatty liver disease in mice [17]. As SGLT2 inhibitors dapagliflozin and empagliflozin improved liver enzymes and decreased liver fat [18] we analyzed also SGLT2 protein expression levels in our NAFLD group to elucidate the factors in the development of NASH.

## 2. Materials and Methods

### 2.1. Patients

Our study was conducted with 55 morbidly obese patients (37 females, 18 males) who have undergone bariatric surgery at a bariatric surgery center. The study group was composed of 22 patients with steatosis (NAFL) and 33 with nonalcoholic steatohepatitis (NASH). Indication for bariatric surgery was based on National Institutes of Health (NIH) guidelines (BMI ≥ 40 kg/m^2^ or ≥35 kg/m^2^, plus co-morbidities) as described before [19]. In addition, the patient selection criteria were the same as described in our previous study [19]. Patients reporting excessive alcohol consumption (>20 g/day in males or >10 g/day in females) indicating alcoholic liver disease were excluded. The surgeon’s choice—i.e., adjustable gastric band, Roux-Y, or gastric bypass surgery—was based on the current guidelines as adapted to the patient’s clinical conditions and comorbidities as well as on clinical experience. Wedge liver biopsies were taken during the procedure. 

There has been conflicting debate about the diagnostic criteria for NASH [20]. The degree of NAFLD can be quantified using the NAFLD activity score (NAS) according to Kleiner et al., where a NAS score ≥5 is defined as NASH [21], or according to the fatty liver inhibition and progression (FLIP) algorithm described by Bedossa et al. [22]. The FLIP algorithm was used by us to classify liver damage in morbid obesity because it allows a more accurate distinction between NAFL and NASH. By using the histologic features of “steatosis”, “ballooning of hepatocytes”, and “inflammation”, the slides were classified as “NAFL” or “NASH”. The HE-stained slides were assessed by two observers (HAB and JK) and the degree of NAFLD was quantified according to the FLIP algorithm of Bedossa et al. [22]. More detailed information on the characteristics of the patients is given in Table 1.

Individual patients’ liver samples were obtained from the files of the Institute of Pathology of the University Hospital of Essen. For all cases, standardized prepared formalin-fixed and paraffin-embedded (FFPE) material was stained with HE, and immunohistochemical staining was performed according to institutional standards. Paraffin-embedded tissue was available in all cases and we reviewed all of them. Informed consent was obtained from every patient. The study was in accordance with the Helsinki Declaration of 1975 and approved by the Ethics Committee of the University Hospital Essen (reference number: 09-4252). 

### 2.2. Histology and Immunohistochemistry

Tissue sections (1 to 2 μm thick) from formalin-fixed and paraffin-embedded (FFPE) tissue blocks were cut, dewaxed, and pretreated. The expression of selected candidate proteins was analyzed by immunohistochemistry, as described previously [23], with an automated staining device (Dako Autostainer, Dako, Glostrup, Denmark). The antibodies used were: anti-Vimentin (#M0725, Dako, Glostrup, Denmark; diluted 1:500 for 60 min at RT); anti-active Caspase 3 (#9661, Cell Signaling, Danvers, MA, USA; diluted 1:50 for 60 min at RT); ki67 (#5278384001, Roche Ventana, Tucson, AZ, USA; undiluted for 60 min at RT); anti-IRF3 (#712217, Invitrogen, San Diego, CA, USA; diluted 1:50 for 30 min at RT); anti-RIG1 (#PA5-110297, Invitrogen; diluted 1:200 for 30 min at RT); anti-M30 (#10700, TecoMedical, Sissach, Switzerland; diluted 1:4500 for 30 min at RT); anti-pSTAT3 (#9145, Cell Signaling; diluted 1:50 for 60 min at RT); anti-pNRF2 (#NBP2-67465, Novus, Centennial, CO, USA; diluted 1:25 for 60 min at RT); anti-syntaxin (#HPA001204, Sigma-Aldrich, Steinheim, Germany; diluted 1:200 for 60 min at RT); anti-SGLT2 (#NBP1-92384, Novus; diluted 1:50 for 30 min at RT); and anti-SOCS3 (#ab280884, Abcam, Cambridge, UK; diluted 1:100 for 30 min at RT). Detection of antigen–antibody binding was performed for vimentin, active Caspase 3, ki67, M30, pSTAT3, syntaxin, SGLT2 and SOCS3 with the ZytoChem Plus AP Polymer Kit (#POLAP-100, Zytomed, Berlin, Germany), and for IRF3, RIG1, and pNRF2 using POLYVIEW^®^ PLUS AP anti-rabbit reagent Rb (#ENZ-ACC110-0150, Enzo Life Sciences, Lörrach, Germany) according to the manufacturer’s protocols. Detailed information on staining protocols is given in Appendix A. Negative controls were included in every run and incubated with non-immune immunoglobulin in the same concentrations but instead of the primary antibody. 

### 2.3. Sample and Immunohistochemistry Evaluation

Immunohistochemical staining was examined with manual IHC scoring and computer-assisted quantification with Aperio ImageScope depending on the protein analyzed. Vimentin, IRF3, M30, and Caspase 3 stainings were studied by visual IHC scoring. Briefly, two independent observers (M.A. and S.S.), blinded to the subgroups of the study, assessed the IHC stains using a semi-quantitative scoring system, analogous to the immunoreactivity scores (IRS) established by Remmele and Stegner [24], as described in our previous studies [25,26]. Vimentin IHC staining was carried out to assess the quality of the FFPE material. The use of vimentin is suitable to monitor the quality of antigen preservation and the uniformity of tissue fixation in FFPE tissues as the epitope of vimentin is partially susceptible to formaldehyde fixation [27].

In assessing IRF3 staining, we counted the number of bile ducts that showed IRF3 immunopositivity. The intensity of IRF3 staining was classified (weak: 1 point; moderate: 2 points; marked: 3 points) and in case of disagreement between the scores, a third observer was consulted. A study of apoptosis was performed by immunohistochemical staining analyses of the antibodies active Caspase 3 and M30. Active Caspase 3 detects endogenous levels of the large fragment of activated Caspase 3 deriving from cleavage adjacent to Asp175. We counted the number of hepatocytes showing positive cytoplasmic staining for cleaved Caspase 3 in each case in the whole slide. Additionally, we quantified apoptosis with the M30 antibody; M30 detects a neo-epitope on caspase-cleaved cytokeratin 18 (CK18) in apoptotic cells; uncleaved CK18 is not detected [28]. Analysis of M30 immunopositivity was performed by scoring the whole slide of each case on apoptotic cells showing intense red cytoplasmatic staining for M30 and counting them. 

We examined the remaining immunohistochemical stains by computer-based automated quantitative IHC scoring. First, stained slides were digitized at 20× resolution using the Aperio AT2 all-slide scanner (Leica, Wetzlar, Germany). The quantification of nuclear stainings (Ki67, pSTAT3, and pNRF2) and cytoplasmic stainings (SOCS3, RIG1, syntaxin, and SGLT2) were performed by Leica image analysis software (Aperio ImageScope). Expression of Ki67, pSTAT3, and pNRF2 was quantified with Aperio nuclear algorithm and expression of SOCS3, RIG1, SGLT2, and syntaxin with Aperio’s positive pixel count algorithm. The percentage of positive hepatocyte nuclei in relation to all hepatocyte nuclei and the percentage of positively stained cytoplasm in relation to the whole area was calculated. 

### 2.4. Statistics

Analyses were carried out with the Statistical Package for Social Sciences (SPSS 28.0, Chicago, IL, USA). We used Mann–Whitney U test for continuous factors and two-sided Fisher’s exact test categorical parameters to analyze associations between clinical/laboratory data and the study groups (NASH/NAFL). Using Mann–Whitney U test and Kruskal–Wallis tests we assessed the association between clinical/laboratory data and the immunopositivity of the proteins. We performed multivariable binary logistic regression analysis with NAFL/NASH as a dependent category and pNRF2, RIG1, and SOCS3 protein expression levels along with clinical and laboratory markers as independent covariates. This aimed to better assess the extent to which the presence of certain clinical and laboratory markers may influence the occurrence of certain protein expressions in NASH/NAFL. Additionally, Pearson’s correlation coefficient was used to evaluate correlations between the protein expressions. All data are shown as medians with ranges presented in parentheses, if not stated otherwise; *p* ≤ 0.05 was defined as statistically significant.

## 3. Results

### 3.1. Higher Expression of RIG1, pNRF2, and SOCS3 in NASH vs. NAFL

Our IHC studies demonstrated that several proteins involved in the innate immune system were differently expressed in NASH and NAFL, which suggests that they play a role in the progression of NAFL to NASH. We examined the percentage of RIG1 immunostained tissue compared to total tissue in all samples and detected cytoplasmic RIG1 expression (Figure 2A–C), whereby the percentage of RIG1 immunopositivity was in NASH (Figure 2C) significantly higher than in NAFL (Figure 2B), shown in boxplots (*p* = 0.01; Figure 2A). Further, to investigate NRF2 activity, we used the NRF2 [p Serine 40] antibody as phosphorylation at the serine 40 residue is required for the dissociation of NRF2 from KEAP1 and the transcriptional activation activity of NRF2 [29,30]. We found that the percentage of pNRF2 immunostained nuclei was in NASH (Figure 2F) significantly higher than in NAFL (Figure 2E), shown in boxplots (*p* = 0.001; Figure 2D). Additionally, the JAK-STAT pathway was activated as cytoplasmic SOCS3 immunostaining was significantly stronger in NASH than in NAFL (*p* < 0.001; Figure 2G–I). However, the comparison between the percentage of pSTAT3-immunostained nuclei in NASH and NAFL showed only a trend (*p* = 0.059; Appendix A). 

### 3.2. Stronger IRF3 Immunostaining of Bile Ducts in NASH Than in NAFL

We investigated the bile ducts regarding their immunostainings for IRF3 and found out that the number of bile ducts with strong immunopositivity for IRF3, namely staining intensity of +3, was in the NASH patients significantly higher than in the NAFL group (*p* = 0.027; Figure 3A,B,D). Further, the average IRF3 staining intensity of the bile ducts was in NASH significantly higher than in NAFL (*p* = 0.002; Figure 3B–D), suggesting that the bile ducts play a role in the innate immune response.

### 3.3. Association between RIG1, pNRF2, SOCS3, IRF3 Immunopositivity and Histological/Laboratory Parameters 

Fasting blood glucose values of patients were divided into three levels: normal range, 70–99 mg/dL; pre-diabetes, 100–125 mg/dL; and diabetes, ≥126 mg/dL according to the criteria of the American Diabetes Association (ADA) [31]. Then, the patients were classified into two groups: we combined the patients with pre-diabetes and diabetes into one group, and the second group aggregated all the patients with normal glucose levels. We detected a significantly higher NRF2 activity in the pre-diabetes and diabetes group than in the normal group (*p* = 0.029; Table 2). Further, we found that patients with steatosis grades 2 and 3 showed significantly higher pNRF2, SOCS3, and RIG1 expression levels than patients with steatosis grades 0 and 1 (*p* = 0.011; *p* = 0.008; *p* = 0.032; Table 2). Regarding ballooning grade, we observed that patients with grade 1 and grade 2 had significantly higher pNRF2 (*p* < 0.001) and SOCS3 (*p* = 0.002; *p* < 0.001; Table 2) protein expression levels than patients with ballooning grade 0. For RIG1 immunopositivity, only patients with ballooning grade 2 had significantly higher RIG1 expression levels than patients with ballooning grade 0 (*p* = 0.010; Table 2). We detected in patients with lobular inflammation 1 higher NRF2 activity than in patients with lobular inflammation 0 (*p* = 0.022; Table 2). Regarding SOCS3, patients with lobular inflammation 1, 2, and 3 had higher SOCS3 protein expressions than patients with lobular inflammation grade 0 (*p* = 0.007; *p* = 0.007; *p* = 0.035; Table 2). Association studies between IRF3 immunopositivity of the bile ducts and clinical/laboratory parameters showed as the only significant result that patients with lobular inflammation grade 2 had significantly stronger IRF3 immunopositivity of the bile ducts than patients with grade 0 (*p* = 0.017).

### 3.4. Multivariable Binary Logistic Regression Analysis

We performed multivariable binary logistic regression analysis for the proteins pNRF2, SOCS3, and RIG1 (Table 3). We found that pNRF2 associated with NASH (*p* = 0.046) was independent of the presence of steatosis (Table 3(A); *p* = 0.01) and lobular inflammation (*p* = 0.016). Further, the multivariable binary logistic regression study showed that RIG1 associated with NASH (Table 3(B); *p* = 0.045) was independent of the presence of steatosis (Table 3(B); *p* = 0.017) and lobular inflammation (Table 3; *p* = 0.009). While RIG1 is significantly associated with NASH shown above by Mann–Whitney U test (Figure 2A; *p* = 0.01), the multivariable binary logistic regression analysis showed that this effect was not seen after adjusting for the variables fasting glucose and total cholesterol (Table 3(C); *p* = 0.285). We detected no associations for SOCS3 protein expression levels along with clinical and laboratory markers as independent covariates by logistic regression studies. Thus, the higher SOCS3 expression levels in NASH are not affected by other clinical and laboratory markers. 

### 3.5. Analysis of Correlations between NRF2 Activation and Syntaxin, Ki67, M30 and SOCS3 Protein Levels

We detected a significant positive correlation of pNRF2 with syntaxin protein expression levels (r = 0.604; *p* < 0.001; Figure 4). Additionally, we found positive associations between pNRF2 and Ki67 (r = 0.645; *p* < 0.001; Figure 4) and M30 (r = 0.431; *p* = 0.001; Figure 4) and SOCS3 (r = 0.560; *p* < 0.001; Figure 4) protein expression levels. 

### 3.6. Analysis of Correlations between SOCS3 and RIG1, Ki67, Syntaxin and SGLT2 Protein Levels

Pearson correlations showed significant positive associations between the protein levels of SOCS3 and RIG1 (r = 0.765; *p* < 0.001; Figure 5), Ki67 (r = 0.463; *p* < 0.001; Figure 5), syntaxin (r = 0.717; *p* < 0.001; Figure 5), and SGLT2 (r = 0.333; *p* = 0.013; Figure 5). 

### 3.7. Analysis of Correlations between IRF3 and pNRF2 and SOCS3 Protein Levels

We found significant Pearson correlations between the IRF3 protein expression in the bile ducts of our patients and the pNRF2 immunopositivity in the nuclei (r = 0.364; *p* = 0.006; Figure 6). Additionally, IRF3 immunostaining of the bile ducts correlated positively with the SOCS3 immunopositivity of the cytoplasm (r = 0.368; *p* = 0.006; Figure 6).

## 4. Discussion

In our present IHC study of NAFLD patients, we detected significantly higher RIG1, IRF3, pNRF2, and SOCS3 protein expressions in NASH patients compared with NAFL patients. These proteins are mainly involved in innate immune regulation [1]; dysregulation of innate immunity plays a role in the pathogenesis of NASH [32]. Further, these proteins were significantly associated with some of the clinical/laboratory parameters and with proteins involved in several biological processes such as autophagy, cell proliferation, and apoptosis. As fatty liver disease is a rising problem worldwide, it is essential to evaluate the mechanisms leading from simple steatosis to advanced non-alcoholic steatosis (NASH). In the following, we discuss how the differentially expressed proteins might be involved in the progression from NAFL to NASH. Based on our results, we assume signaling cascades activated by PRRs, FFA accumulation, and ROS formation (for more details see Figure 1).

RIG1 protein, belonging to the pattern recognition receptors (PRR) family, was significantly higher in NASH patients than in those with NAFL. It was reported that the levels of another PRR protein, namely STING (stimulator of interferon genes) 1, were increased in NASH patients [33,34]. It is documented that recognition of extracellular pathogens or endogenous injury signaling by PRRs leads to signal transmission to downstream MAPK signal cascades, resulting in the activation of ERK/JNK/p38MAPK/NF-κB signals and the transcription of proinflammatory cytokine genes and type I/II IFNs. Intriguingly, PRRs participate in the regulation of lipid metabolism [1]. Fatty acid toxicity is known to trigger the initiation and continuous activation of an inflammatory response in the liver of NAFLD patients [35]. Thus, our detection of higher RIG1 (a PRR) expression in NASH vs. NAFL is in line with the literature. There are only a few study groups analyzing RIG-1 expression immunohistochemically in patients; Frietze et al. analyzed 12 NASH patients and 5 normal controls and found a reduction in RIG-1 protein expression in NASH compared to normal controls. Nevertheless, their patient cohort was very small, and they did not compare NASH expression with NAFL as we did but with normal controls. 

Additionally, we found that bile ducts showed stronger IRF3 immunostaining in NASH than in NAFL. We also detected that patients with lobular inflammation grade 2 showed higher IRF3 immunopositivity regarding bile ducts than patients with grade 0 (*p* = 0.017). This is in line with the literature as RIG-1 interacts with dsRNA and initiates downstream signaling which leads to IRF3 and NF-κB activation [36]. Expression of IRF3 in bile ducts is also reported in the literature [37,38,39]. It is documented that biliary innate immunity is implicated in the pathogenesis of various cholangiopathies in biliary tract diseases and biliary tract defense systems [38]. Cholangiocytes have an innate immune system and the biliary epithelial cells express a variety of PRRs such as Toll-like receptors (TLRs) and they recognize both bacterial and viral PAMPs. Cultured biliary epithelial cells were able to recognize viral PAMPs such as double-stranded RNA (dsRNA). After stimulation with poly(I:C), (analog of viral dsRNA), cultured human biliary epithelial cells expressed NF-κB and IRF3 [37,38]. Shimada performed experiments with human biliary epithelial HuCCT1 cells. They observed that TLR3 signaling led to the expression of CCL5 via NF-κB and IRF3 in bile duct cells, and they suggested the involvement of this signaling pathway in biliary atresia pathogenesis. The involvement of bile duct cells in the development of nonalcoholic fatty liver disease was reported; however, they did not study IRF3 expression but cellular senescence markers and chemokines [40].

NRF2 is described in the literature as a “double-edged sword” [41]. Under normal conditions, NRF2 is mainly localized in the cytoplasm attached to Keap1, which causes inhibition of NRF2 activity because of proteasomal degradation of NRF2. Under stressed conditions, NRF2 dissociates from Keap1, translocates to the nucleus, and promotes the expression of antioxidant response element (ARE) genes, which control stress response, antioxidant defense, drug metabolism, proteasomal degradation, and cell proliferation [8,42]. On the other side, NRF2 can function as a proto-oncogene promoting the growth of cancer cells [43]; liver tumorigenesis by NRF2 activation was reported [44]. Autophagy impairment and NRF2 activation have induced chaperone-mediated autophagy activation and tumor cell survival [45]. We detected higher NRF2 activity in NASH versus NAFL; our result is consistent with aberrant NRF2 activation in various cancer cells and cancer tissues [46] driven by different causes of NRF2 activation. In hepatocellular Huh1 carcinoma cell lines phosphorylated p62 accumulation was documented as the reason for NRF2 activation. This is in accordance with another IHC study of patients with chronic liver disease [41]. Mohs et al. reported an association of NRF2 activity in patients with the grade of inflammation, which we also observed. In contrast, they did not detect an association with steatosis, as we did. It is reported that age can explain different results found in the literature regarding the effects of NRF2 activation. In young animals, a protective role of NRF2 in the development of hepatitis and steatosis has been observed [5]. On the contrary, in older mice, NRF2 activation has induced the activation of genes involved in lipid synthesis and uptake [5]. Since we found a trend toward higher NRF2 activation in older patients, this is consistent to some extent with our results. The reasons for NRF2 activation in tumor tissue can be mutations in KEAP-1 [47] causing Keap-1 not to be able to bind to NRF2, resulting in constitutive NRF2 activation, increased ROS formation because of increased metabolism of fatty acids [5], and increased influx of unfolded proteins into the endoplasmatic reticulum (ER) inducing ER-stress [48]. Also, a noncanonical mechanism of NRF2 activation by autophagy deficiency is documented [49]. We found higher glucose levels in NASH patients; this is consistent with the literature, as those in transgenic mice with enhanced NRF2 activity blood glucose levels were elevated [50]. Also, Islam et al. documented the association of NRF2 activity with glucose metabolism [51]. He et al. [44] reported that NRF2 activation alters glucose and lipid metabolism; hepatocyte-specific NRF2 activation in a mouse model, caused by accumulation of p62 or inhibition of KEAP1 binding, resulted in hepatomegaly associated with increased glycogenosis, steatosis, and G2/M cell cycle arrest, favoring hyperplasia without cell division. Additionally, we detected that patients with ballooning grades 2 and 1 had higher pNRF2 levels than patients with grade 0. It is known that in ferroptosis, an iron-dependent, lipid peroxidation-driven cell death cascade, the formation of a “ballooning” phenotype describes its final critical feature; many of the key anti-ferroptotic pathway components are under the transcriptional control of NRF2 [52]. Dodson et al. studied the association between NRF2 activity and ferroptosis and found that NRF2 plays a critical role in attenuating lipid peroxidation and ferroptosis [52]. Interestingly, the double-sword effect of NRF2 activation is also observed in clinical trials: a phase 3 clinical trial of bardoxolone methyl (activator of the NRF2 pathway) for the treatment of type 2 diabetes and stage 4 chronic kidney disease was discontinued because it failed to reduce the risk of end-stage renal disease/death [53].

Further, we detected significantly higher SOCS3 expression in the cytoplasm of NASH patients than in NAFL. Usually, transcription of SOCS genes is activated following stimulation with cytokines; in the SOCS3 promoters STAT-binding sites were detected; transfection with dominant-negative STAT3 inhibits cytokine-induced expression of SOCS3 [54]. STAT3 overexpression and constitutive activation have been commonly recorded in HCC and are associated with poor prognosis [2]. Kim et al. found in high-fat diet-fed rat livers higher SOCS3 protein expressions than in normal diet-fed rat livers, which is in line with our results [55]. Zhang et al. detected that fructose-treated mouse cells had higher STAT3 and SOCS3 levels [56]. Fructose treatment-induced inflammation activity which was associated with lipid accumulation. We also found a tendential upregulation of pSTAT3 in NASH versus NAFL (*p* = 0.059) which is in accordance with the literature as SOCS3 is a target gene of STAT3 [57]. On the other side, as the association regarding STAT3 is only tendential, we suggest that additional other factors than STAT3 contribute to the activation of SOCS3. It is documented that the expression of the SOCS proteins is increased by cytokine signaling through the activation of STAT- and NF-KB-mediated pathways [58]. Ueki demonstrated in a mouse model that in both obesity and lipopolysaccharide (LPS)-induced endotoxemia there is an increase in SOCS proteins, SOCS1 and SOCS3, in the liver. Senn et al. observed that IL-6 treatment of mice leads to SOCS3 expression in the livers and also inhibited hepatic insulin receptor signaling in these animals [59]. We did not find a significant association between fasting glucose levels and SOCS3 protein expression. Nevertheless, the upregulation of SOCS3 in our patients can be caused by other factors than IL6 levels. It should be noted that few animal models faithfully replicate human disease. Bi et al. showed that in hepatocytes steatosis was alleviated by reducing SOCS3 by inhibiting JAK2/STAT3 pathway [60]. This is also in accordance with our results as we detected significant positive associations between enhanced SOCS3 expression and steatosis grade. Handa et al. observed increased gene expression of STAT3 in NASH patients versus NAFL. However, SOCS3 expression was significantly reduced in patients with NASH which is contradictory to our results [61]. They discussed why they did not observe upregulation of SOCS3, a known negative regulator of the JAK2/STAT3 signaling pathway in their cohort, and suggested that other mechanisms distinct from SOCS3 were involved in the suppression of inflammatory response. Ogata et al. studied SOCS3 in human HCC and detected that SOCS3 expression levels were significantly higher in non-HCC regions of the liver in 20 HCV-infected patients than in 17 non–HCV-infected patients [62]. However, their results were related to HCC patients and not NAFLD, and the cohort of patients studied was smaller than ours. Sharma et al. reported that SOCS3 expression was increased by adiponectin [63]. Bechmann, Canbay et al. studied NASH patients and reported that serum FFA, BA, and M30 were increased in NASH compared with simple steatosis, while adiponectin was significantly decreased [64]. In our patient cohort, we found no significant differences regarding adiponectin levels between the NASH and NAFL patients. However, the literature shows that adiponectin should not be considered only as a “good cytokine” and that its role is much more complex. Guo et al. reported that adiponectin knockout causes a protective effect against high-fat diet-induced liver injury, possibly related to autophagy regulation, despite persistent liver steatosis [65]. However, the reasons why we did not detect lower adiponectin in NASH may be that data referring to adiponectin was not available for all cases and, secondly, in our cohort, the number of females was relatively high. It is known that adiponectin is higher in women than in men [66], so this may also have an effect. We observed in two NASH cases adiponectin serum levels of about 8 and 11; we suppose that these patients had a more favorable disease course. Unfortunately, we cannot verify this as follow-up data is lacking.

We performed correlation studies to understand how the proteins upregulated in NASH are in association with each other and with autophagy, cell proliferation, and apoptosis markers. Upregulation of pNRF2 was positively correlated with the autophagy marker protein Syntaxin. In NASH patients, free fatty acids (FFA) are known to be elevated and correlate positively with disease severity [67]. FFAs induce ER stress response [68] leading to NRF2 activation and upregulation of NRF2 target genes, several of these genes induce autophagy [69]. Also, other studies described that the NRF2–KEAP1 pathway provides a positive feedback loop for autophagy activation. Also, the unfolded protein response (UPR) induced by ER stress leads to the activation of autophagy [70].

We detected a positive correlation between NRF2 activation and the cell proliferation marker Ki67 and, simultaneously, a positive correlation with apoptotic cleavage of CK18 indicated by the apoptosis marker M30. These results appear contradictory at first sight. The correlation between pNRF2 and cell proliferation (r = 0.645; *p* > 0.000) was stronger than the correlation between pNRF2 and M30 (r = 0.431; *p* = 0.001). In mouse hepatoma (Hepa-1) and human hepatoblastoma (HepG2) cells, NRF2 activation upregulated the antiapoptotic protein Bcl-2, prevented apoptosis, and increased tumor cell survival and growth/proliferation [71]. On the other side in transgenic mice, activated NRF2 delayed cell proliferation and enhanced the apoptosis of damaged liver cells [72]; apoptotic response after NRF2 activation by ROS is reported [73]. It is known that NRF2 activation causes cancer prevention and or progression; this depends on the cellular context and environment [74]. We consider that different biological processes take place in the organism in parallel. In addition, even if the patients are classified into a specific entity regarding NAFLD according to Bedossa criteria, each patient case represents a unique profile.

Further, we found a positive correlation between NRF2 activation and SOCS3 protein expression. This is in accordance with the literature as Meng et al. demonstrated that the p-STAT3/SOCS3 pathway and the KEAP1/NRF2 pathways are linked. They demonstrated that SOCS3 can directly bind to KEAP1 preventing the degradation of NRF2 and resulting in NRF2 activation [75]. We detected a positive correlation between SOCS3 and RIG1 which is consistent with the literature as the pattern recognition receptor RIG1 can lead to an activation of JAK-STAT signaling [1]. In our patient cohort, SOCS3 was positively correlated with the autophagy marker syntaxin. This is in accordance with the literature as Wan et al. showed a positive association between autophagy activation and upregulation of SOCS3. Briefly, AMPK-autophagy activation suppressed neuroinflammation and improved morphine tolerance via the upregulation of SOCS3 by inhibiting miRNA-30a-5p [76]. They also underlined the dual role of autophagy in tumor development: autophagy prevents tumor initiation in early tumorigenesis but once the tumor progresses autophagy contributes to tumor survival [76]. Further, we performed IHC staining for SGLT2 and detected immunopositive staining in the cytoplasm of hepatocytes which is in line with the literature [77]. We did not find significant differences regarding SGLT2 protein expression between NASH and NAFL. Nakano et al. also observed no significant difference in hepatic expression of SGLT2 in the stratified analysis according to age, sex, BMI, and the severity of the liver disease. We detected a positive correlation between SGLT2 and SOCS3 expression levels concomitant with the literature, as SOCS3 is a mediator of insulin resistance in the liver [78].

## 5. Conclusions

Our results demonstrate significantly higher expression of NRF2 and SOCS3 proteins in our NASH patients than in those with NAFL. We suggest that they are most likely promising candidates for further studies to detect drugs for therapy. The relationship between fasting glucose levels and NRF2 is of interest regarding metabolic syndrome therapy. We recommend immunohistochemical staining of liver tissue for pNRF2 during liver biopsy to detect prediabetic abnormality in advance. The association of SOCS3 with autophagy and SGLT2 expression are most certainly interesting and potential areas for further studies, such as the drug combination of SGLT2 inhibitors and SOCS3 modulators.

## Figures and Tables

**Figure 1 jpm-13-01152-f001:**
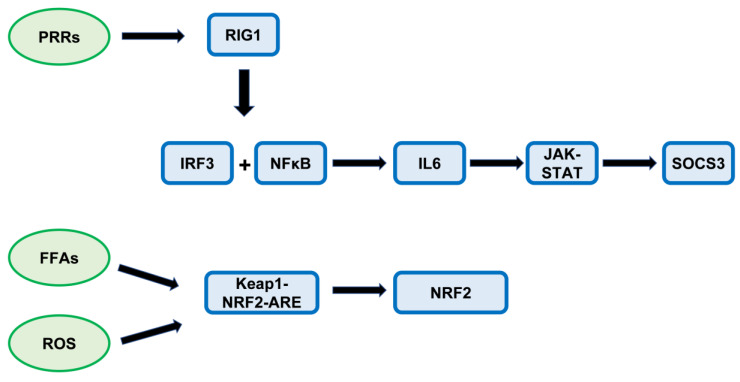
Summary figure representing the signaling cascades. We suppose that activation by PRRs triggers RIG1 expression. In addition, FFA-accumulation, and ROS formation leads to Keap1-NRF2-ARE activation, whereby it is likely that many biological processes take place in the organism in a parallel manner (for example NFκB activation is known to induce ROS formation). Abbreviations: PRRs: pattern recognition receptors; FFAs: free fatty acids; ROS: reactive oxygen species; Keap1: Kelch-like ECH-associated protein 1; ARE: antioxidant response elements.

**Figure 2 jpm-13-01152-f002:**
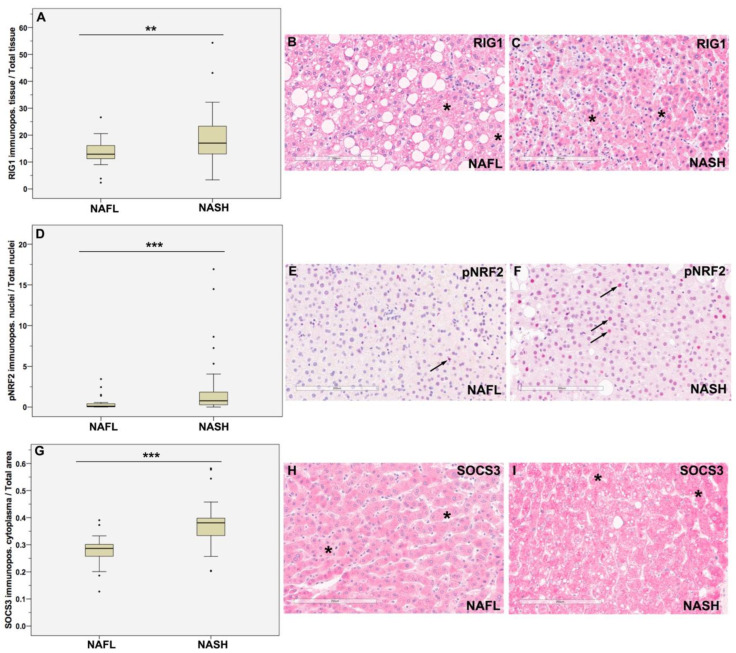
Semi-quantitative immunohistochemical analysis of RIG-1, pNRF2, and SOCS3 and protein expressions. Images show representative cases of our NAFLD cohort. (**A**–**C**) We detected stronger cytoplasmic RIG1 expression in NASH than in NAFL. (**A**–**C**) Boxplots depict that the percentage of RIG1 stained tissue to total tissue is higher in NASH than in NAFL; ** *p* = 0.01. (**A**) In the IHC images, representative cytoplasmic RIG1 staining in NAFL (**B**) and NASH (**C**) is shown (asterisk). Further, the ratio of pNRF2 stained nuclei to total nuclei was higher in NASH than in NAFL; *** *p* = 0.001 (**D**). Representative pNRF2 IHC images of NAFL (**E**) and NASH (**F**) demonstrate stronger nuclear staining (arrows) in NASH than in NAFL. Also, cytoplasmic SOCS3 expression was stronger in NASH than in NAFL (**G**–**I**). Boxplots depict the ratio of the area with cytoplasmic SOCS3 immunopositivity to the total area with increased SOCS3 expression in NASH compared with NAFL; *** *p* < 0.001 (**G**). Representative SOCS3 IHC images of NAFL (**H**) and NASH (**I**) show representative areas of cytoplasmic SOCS3 immunopositivity (asterisk). Differences between groups were analyzed using Mann–Whitney U tests; bold lines inside the box plot represent median levels. Results are significant at * *p* ≤ 0.05, ** *p* ≤ 0.01 and *** *p* ≤ 0.001; bars = 200 μm.

**Figure 3 jpm-13-01152-f003:**
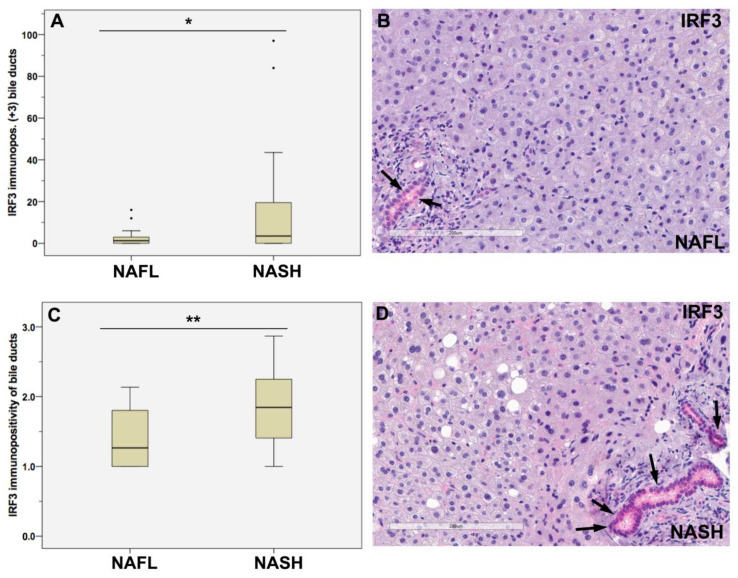
Associations between IRF3 immunostainings and disease progression. Study of the bile ducts in the cohort depicted that IRF3 staining intensity was in NASH higher than in NAFL (**A**–**D**). In NASH, there are more bile ducts having a strong IRF3 staining intensity of +3 than in NAFL; * *p* = 0.027 (**A**). The average IRF3 staining intensity of the bile ducts per case was in NASH higher than in NAFL; *p* = 0.002 (**C**). Representative IHC images of NAFL (**B**) and NASH (**D**) show representative areas of IRF3 immunopositive stained bile ducts (arrows) in NAFL (**B**) and NASH (**D**) with stronger IRF3 staining in NASH. Differences between groups were analyzed using Mann–Whitney U tests; bold lines inside the box plot represent median levels. Results are significant at * *p* ≤ 0.05 and ** *p* ≤ 0.01; bars = 200 μm.

**Figure 4 jpm-13-01152-f004:**
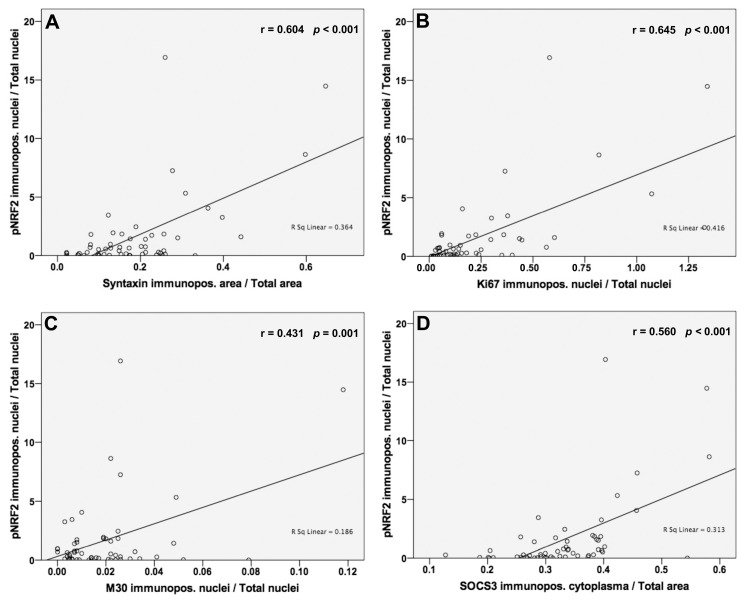
pNRF2 protein expressions are positively associated with RIG1, syntaxin, Ki67, M30, and SOCS3 expressions. (**A**,**B**) We observed a high degree of correlations between pNRF2 and syntaxin (r = 0.604; *p* < 0.001) and Ki67 (r = 0.645; *p* < 0.001). (**C**,**D**) The degree of correlation between pNRF2 and M30 was moderate (r = 0.431; *p* = 0.001) and between pNRF2 and SOCS3 was high (r = 0.560; *p* < 0.001). Pearson’s correlation coefficient was used to evaluate protein expression results with significance set at *p* ≤ 0.05. High degree of correlation: Pearson’s correlation coefficient, also known as Pearson’s r = 0.5–1.0; moderate degree of correlation: Pearson’s r = 0.3–0.49; low degree of correlation: Pearson’s r < 0.3.

**Figure 5 jpm-13-01152-f005:**
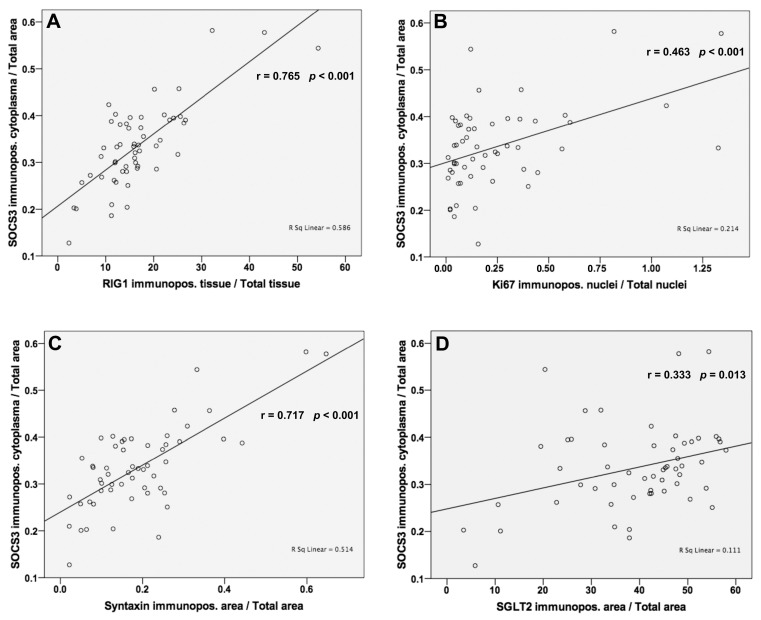
SOCS3 protein expressions are positively associated with RIG1, Ki67, syntaxin, and SGLT2 expressions. (**A**,**C**) High degree of correlations were detected to RIG1 (r = 0.765; *p* < 0.001) and syntaxin (r = 0.717; *p* < 0.001). (**B**,**D**) The degree of correlations to Ki67 and to SGLT2 were significant but moderate (r = 0.463; *p* < 0.001 and r = 0.333; *p* = 0.010). Pearson’s correlation coefficient was used to evaluate protein expression results with significance set at *p* ≤ 0.05.

**Figure 6 jpm-13-01152-f006:**
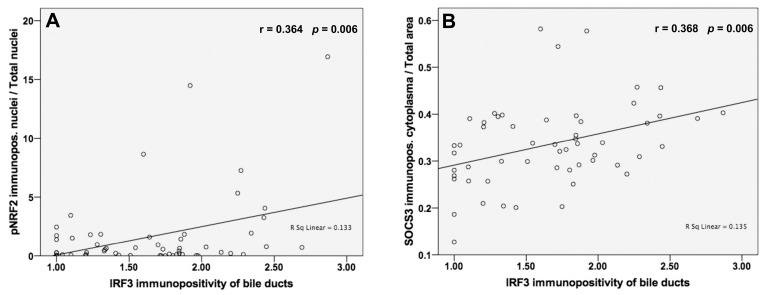
IRF3 immunostaining of the bile ducts is associated with NRF2 activity and with SOCS3 protein expressions. (**A**) We depicted positive significant correlations of moderate degree to pNRF2 immunopositivity of the nuclei (r = 0.364; *p* = 0.006) and (**B**) SOCS3 cytoplasmic staining (r = 0.368; *p* = 0.006). Pearson’s correlation coefficient was performed with significance set at *p* ≤ 0.05.

**Table 1 jpm-13-01152-t001:** Clinical and laboratory data of the study groups.

Characteristics	NAFL (*n* = 22)	NASH (*n* = 33)	*n* Valid Cases	*p* Value *
Age (years)	38 (24–67)	45 (20–67)	22/33	0.110
Gender (male/female)	4/18	14/19	22/33	0.082
BMI (kg/m^2^)	49.9 (29.4–66.9)	53 (27.4–78.19)	21/31	0.208
Adiponectin (µg/mL)	3.35 (1.3–8.28)	2.87 (0.83–11.9)	20/24	0.759
CK18 M30 (IU/L)	174.7 (61.8–807.7)	366.6 (80.1–1573.9)	20/24	0.002
CK18 M65 (IU/L)	331.9 (87.9–960.1)	628.6 (255.8–5273.1)	19/24	<0.001
Fasting Glucose (mg/dL)	95.50 (73–150)	120 (72–385)	22/29	0.001
Total Cholesterol (mg/dL)	198 (120–261)	177.5 (116–247)	15/18	0.320
Triglyceride (mg/dL)	149 (34–218)	207 (55–421)	13/12	0.041
ALT (U/L)	20 (13–65)	39 (14–120)	22/31	0.001
AST (U/L)	23 (16–49)	32.5 (23–90)	14/20	<0.001
GGT (U/L)	20 (2–93)	43 (15–1213)	22/31	<0.001
Fibrosis Grade				
(0 + 1)	4 + 7	6 + 7		
(2 + 3)	10 + 1	18 + 2	22/33	0.58
Steatosis Grade				
(0 + 1)	0 + 19	0 + 7		
(2 + 3)	3 + 0	14 + 12	22/33	<0.001
Ballooning Grade				
(0 + 1)	17 + 3	0 + 16		
2	2	17	22/33	0.001
Lob. Inflam. Grade				
(0 + 1)	16 + 4	0 + 11		
(2 + 3)	2 + 0	20 + 2	22/33	<0.001

The presented values are medians; ranges are enclosed in parentheses. * *p* values correspond to the comparison of NAFL/NASH and *n* valid cases reports the number of valid NAFL/NASH cases used for the statistical analysis. Regarding fibrosis grade, steatosis grade, ballooning grade, and lobular inflammation grade, we placed the two grades 0 and 1 in a common group. For example, statistical evaluation was performed in the analysis of steatosis grade by comparing the number of NAFL and NASH patients in group 1 (grade 0 + 1) with the number of NAFL and NASH patients in group 2 (grade 2 + 3). We used the statistic test Mann–Whitney U test for continuous factors and two-sided Fisher’s exact test for categorical parameters. *p* ≤ 0.05 was defined as statistically significant. Abbreviations: *n* = number; NAFL = non-alcoholic fatty liver; NASH = non-alcoholic steatohepatitis; ALT = alanine aminotransferase; BMI = body mass index; AST = aspartate aminotransferase; GGT = gamma-glutamyl transferase; CK18 = Cytokeratin18; Lob. Inflam. Grade = lobular inflammation grade.

**Table 2 jpm-13-01152-t002:** Association of clinical/laboratory parameters with IHC studies.

		pNRF2		SOCS3		RIG1	
Parameters	*n* Valid Cases	Median Value ^b^	*p* Value *	MedianValue ^b^	*p* Value *	MedianValue ^b^	*p* Value ***
Fasting Glucose (mg/dL)							
Normal (70–99)	19	0.12 (0–7.25)		0.29 (0.12–0.45)		14.31 (2.33–25.25)	
Pre-diabetes and Diabetes ^a^	32	0.69 (0.01–16.93)	0.029	0.33 (0.18–0.45)	0.090	15.98 (3.38–26.62)	0.205
Total Cholesterol (mg/dL)							
Normal (<200)	20	0.27 (0.01–4.06)		0.3 (0.18–0.45)		14.45 (3.38–26.28)	
Elevated (>200)	13	0.15 (0–7.25)	0.957	0.31 (0.2–0.45)	0.813	12.14 (3.84–26.62)	0.548
Triglyceride (mg/dL)							
Normal (<200)	18	0.06 (0–1.8)		0.3 (0.18–0.39)		12.82 (3.84–23.33)	
Elevated (>200)	7	0.2 (0.01–1.73)	0.297	0.27 (0.2–0.39)	0.701	14.6 (3.38–25.56)	0.495
ALT (U/L)							
Normal (<35/<50)	35	0.29 (0–7.25)		0.31 (0.12–0.45)		14.21 (2.33–26.62)	
Elevated (>35/>50)	18	0.67 (0.02–16.93)	0.195	0.34 (0.2–0.57)	0.276	16.29 (5.02–43.11)	0.244
AST (U/L)							
Normal (<35/<50)	24	0.1 (0–3.45)		0.3 (0.18–0.4)		14.16 (3.38–25.03)	
Elevated (>35/>50)	10	0.57 (0.1–16.93)	0.061	0.34(0.2–0.57)	0.223	14.53 (5.02–43.11)	0.539
GGT (U/L)							
Normal (<35/<55)	36	0.26 (0–16.93)		0.32 (0.12–0.57)		14.26 (2.33–43.11)	
Elevated (>35/>55)	17	0.93 (0.03–5.33)	0.054	0.33 (0.2–0.42)	0.607	16.02 (5.02–26.62)	0.261
Age Grade							
Young (20–40)	28	0.27 (0–14.48)		0.31 (0.2–0.57)		14.32 (3.38–43.11)	
Old (41–67)	27	0.74 (0–16.93)	0.055	0.33 (0.12–0.58)	0.245	16.21 (2.33- 54.32)	0.225
Fibrosis Grade							
0	10	1.4 (0.18–3.45)		0.34 (0.28–0.39)		16.69 (9.59–26.62)	
1	14	0.26 (0.04–7.25)	0.089	0.3 (0.12–0.45)	0.320	15.21 (2.33–25.25)	0.380
2	28	0.34 (0–16.93)	0.091	0.32 (0.2–0.58)	0.619	14.94 (3.38–54.32)	0.691
3	3	0.68 (0.21–2.45)	0.735	0.33 (0.33–0.37)	1.000	12.99 (12.3–17.32)	0.398
Steatosis Grade							
0 + 1	26	0.11 (0–7.25)		0.29 (0.12–0.45)		14.28 (2.33–25.25)	
2 + 3	29	0.77 (0–16.93)	0.011	0.35 (0.2–0.58)	0.008	17.03 (3.38–54.32)	0.032
Ballooning Grade							
0	17	0.05 (0–0.56)		0.28 (0.18–0.37)		12.14 (3.84–20.57)	
1	19	0.93 (0.1–16.93)	<0.001	0.34 (0.12–0.45)	0.002	15.93 (2.33–25.56)	0.068
2	19	1.42 (0–14.48)	<0.001	0.38 (0.2–0.58)	<0.001	17.03 (3.38–54.32)	0.010
Lobular Inflammation Grade							
0	16	0.09 (0–3.45)		0.28 (0.12–0.39)		12.91 (2.33–26.62)	
1	15	0.68 (0.05–7.25)	0.022	0.35 (0.18–0.45)	0.007	16.12 (5.02–26.28)	0.097
2	22	0.66 (0–16.93)	0.051	0.36 (0.2–0.58)	0.007	16.42 (3.38–54.32)	0.174
3	2	1.67 (0.1–3.26)	0.325	0.36 (0.33–0.39)	0.035	15.45 (15.1–15.82)	0.399

^a^ We defined fasting blood glucose level of 100–125 mg/dL as pre-diabetes and glucose levels of ≥126 mg/dL as diabetes. ^b^ Values are the median of pNRF2, SOCS3, and RIG1 immunopositivity with ranges presented in parentheses. ALT and AST threshold for normal values were <35 U/L for females and <50 U/L for males; GGT threshold for normal values were <35 U/L for females and <55 U/L for males. * *p* values correspond to the analysis of the association between clinical/laboratory parameters with pNRF2, SOCS3, and RIG1 immunopositivity in the NAFLD cohort. Mann–Whitney U test and Kruskal–Wallis test were used for statistical analysis of the difference between two or more groups. Results are significant at *p* ≤ 0.05.

**Table 3 jpm-13-01152-t003:** Multivariable binary logistic regression studies.

	Exp (B)	95% CI		*p* Value
		Lower	Upper	
A				
pNRF2	2.456	1.017	5.93	0.046
Age (years)	1.006	0.926	1.092	0.893
BMI (kg/m^2^)	1.102	0.947	1.283	0.208
Fibrosis Grade				
1 vs. 0	1.747	0.072	42.549	0.732
2 vs. 0	1.396	0.112	17.384	0.795
3 vs. 0	0.989	0.026	38.383	0.995
Steatosis Grade				
(2 + 3) vs. (0 + 1)	29.267	2.237	382.895	0.01
Ballooning Grade				
2 vs. (0 + 1)	0.628	0.041	9.674	0.739
Lob. Inflam. Grade				
(2 + 3) vs. (0 + 1)	19.197	1.738	211.979	0.016
B				
RIG1	1.21	1.005	1.457	0.045
Age (years)	1.085	0.997	1.18	0.058
BMI (kg/m^2^)	1.018	0.898	1.153	0.782
Fibrosis Grade				
1 vs. 0	2.994	0.148	60.656	0.475
2 vs. 0	1.885	0.131	27.206	0.642
3 vs. 0	0.843	0.022	32.364	0.927
Steatosis Grade				
(2 + 3) vs. (0 + 1)	24.689	1.788	340.962	0.017
Ballooning Grade				
2 vs. (0 + 1)	0.384	0.021	7.164	0.521
Lob. Inflam. Grade				
(2 + 3) vs. (0 + 1)	35.992	2.454	527.784	0.009
C				
RIG1	1.080	0.938	1.243	0.285
Fasting glucose (mg/dL)	1.052	1.008	1.097	0.020
Total cholesterol (mg/dL)	0.995	0.972	1.018	0.682

Multivariable binary logistic regression analysis with NAFL/NASH as a dependent category and pNRF2 and RIG1 protein expression levels along with clinical and laboratory markers as independent covariates for (A) pNRF2 and (B) RIG1 together with age, BMI, fibrosis, steatosis, ballooning, and lobular inflammation grades as covariates. (C) RIG1 together with fasting glucose and total cholesterol levels as covariates. As the reference for the categorical covariate fibrosis “Grade 0” is chosen, the reference for the categorical covariates steatosis, ballooning, and lobular inflammation is “Grade (0 + 1)”.

## Data Availability

Data is provided in the manuscript and/or Appendix A. The source of the data came from the Institute of Pathology, University Hospital of Essen.

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
