# Peer review of "Higher pNRF2, SOCS3, IRF3, and RIG1 Tissue Protein Expression in NASH Patients versus NAFL Patients: pNRF2 Expression Is Concomitantly Associated with Elevated Fasting Glucose Levels"

_jpm, 2023, doi:10.3390/jpm13071152_

Round 1
Reviewer 1 Report
In this study, the authors aimed to clarify which proteins play a role in the progression of NAFL to NASH. They investigated paraffin-embedded samples of 22 NAFL and 33 NASH patients, and found that the expression of pNRF2 immunopositive nuclei and SOCS3 cytoplasmic staining were higher in NASH than in NAFL (p=0.001); pNRF2 was associated with elevated fasting glucose levels. SOCS3 immuno-positivity correlated positively with RIG1 (r=0.765; p=0.001). Further, in NASH bile ducts showed stronger IRF3 immunostaining than in NAFL (p=0.002); immunopositive RIG1 was higher in NASH than in NAFL (p=0.01), indicating that pNRF2, SOCS3, IRF3, and RIG1 are involved in hepatic lipid metabolism.
Some critiques are as following:
1. In the figure 1, it’s hard to distinguish the results of the Immunohistochemical positive staining.
2. The section of discussion was over discussed.
3. Histology and immunohistochemistry were performed, the semi-quantities of these images were analyzed to demonstrate how the expressions of the proteins were. So, the figure 1 and figure 2 should be merged. The images of the figure 3 about IRF3 immunostaining of bile ducts should be listed separately from Figure 1.
4. Just as what the authors described about the expressions of RIG-1 and NRF-2 or other proteins reported by others study in the MS, which showed some rare evidences in NASH, the correlations between these proteins need to be confirmed in vitro or in vivo experiments.
Reviewer 2 Report
the manuscript is scientifically sounding. the results demonstrate that there is higher pNRF2, SOCS3, IRF3, and RIG1 protein expression in NASH patients versus NAFL patients. pNRF2 expression associated with elevated fasting glucose levels are involved in the pathogenesis of metabolic syndrome.
Minor revisions are needed as follows:
- in methodology, liver samples collected should have more details and please explain how did you get them from patients file as mentioned
- table 1 should be mentioned in results chapter not materials and methods
- IHC should be re-written in a more organized way according to the antibodies mentioned in table S1
- in figure 2 legend, * means P value is exactly equal to 0.01 or equal or less than 0.01. the same is for **
- Figure 6 should be transferred to introduction rather than putting it in discussion chapter
- Recommendation should be added
Reviewer 3 Report
Dear authors,
thank you for your effort, I read the article with interest. And I recommended it for publication with minor revisions.
Title: please add "tissue" in front of protein expression.
Introduction is well written and thorough, however I would like to ask what was the reason behind selection of these particular proteins? (e.g. staining availability, price etc). please add the reason to the introduction.
Methods: Table one, please describe exactly the what the numbers after the slash "/" in lines Fibrosis grade, Statosis grade etc represent.
Results . subheading - association beween RIG1, ... imunopositivity and clinical/lab parameters should be changes to histological/lab parameter, because no clinical parameters /eg. BMI/ are reported.
Results. the difference of protein expression between NASH and NAFL should be also analysed multivariately to account for the presence of confounders (eg BMI, degree of fibrosis etc.) I suggest doing a multivariate logistic regression with NAFL/NASH as a dependent and protein expression categories along with BMI, age fibrosis as independents.
Also the correlations between reported parameters should be controlled for aforementioned confounders
Round 2
Reviewer 1 Report
none